# OpenReview forum: "Implicit Intelligence - Evaluating Agents on What Users Don’t Say"
_ICML.cc/2026/Conference — ICML 2026 regular_

### Official Review · Reviewer_xXLU · 2026-03-08

**Soundness:** 2
**Presentation:** 3
**Significance:** 2
**Originality:** 3
**Overall Recommendation:** 4
**Confidence:** 5

**Summary:**

This paper introduces a benchmark for testing the implicit intelligence of LLM-based agents. To do so, they developed Agent-as-a-World (AaW), in which worlds are defined in YAML files and agents can interact with them. Their results on 16 frontier models show that models often can’t satisfy all implicit user requirements.

**Compliance With Llm Reviewing Policy:**

Affirmed.

**Final Justification:**

I've raised my score from 3 to 4.

After the rebuttal, I'm more convinced that the lack of a real environment could help us remove the noise and focus on the effects.

I still think there's a better version of this paper where the world is created from a web environment, and then the results are validated in the original realistic environment.

**Key Questions For Authors:**

1. Did you consider generating these worlds from realistic websites/apps to make sure that what you learn translates back to realistic setups?
2. Is it fair to expect agents to satisfy all these implicit requirements without communication?

**Limitations:**

Yes, there are limitations. The impact statement is missing; it’s under Acknowledgments.

**Strengths And Weaknesses:**

### Strengths

- The paper is clear and easy to follow.
- The idea of having a framework that can be adapted more easily to fit a particular goal (in this case, implicit intelligence) is interesting and relevant.
- Experiments were run with a lot of frontier models, more than usual.

### Weaknesses

The main weakness of this paper is having to generate artificial environments to test agents. While it makes some things easier, results won’t necessarily generalize to realistic environments where these agents are going to be deployed.

You mention that engineered environments “impose significant overhead; each new domain requires dedicated engineering, and the infrastructure itself becomes a barrier to rapid iteration.” While this is true, there are already a lot of environments that can be adapted. For example, a recent paper [1] accomplishes this by transforming the HTML with interventions before passing it to the agents. That allows them to keep realistic environments while being able to create counterfactuals to test different conditions.

Instead, using your framework in other environments also requires generating these worlds. In the time of vibe coding, one might argue that it’s simpler to generate a website where agents can take actions. Moreover, your environments are quite simple and don’t fully cover the experience of an agent when they need to act in real scenarios. That complexity plays a big role, and without it, we might reach the wrong conclusions.

Finally, the analysis looks at these agents from a functional perspective (i.e., whether they have an ability or not), but nothing in terms of their actions or what leads them to succeed or fail.

[1] Cherep, M., Ma, C., Xu, A., Shaked, M., Maes, P., & Singh, N. (2025). A Framework for Studying AI Agent Behavior: Evidence from Consumer Choice Experiments. arXiv preprint arXiv:2509.25609.

### Other

Most figures are unnecessarily large, and the font is too small

---

> ### Author Rebuttal · Authors · 2026-03-30
>
> We thank Reviewer xXLU for the thoughtful review and engagement with our evaluation design choices. We address each concern below.
>
> **Design Philosophy: Simulated Environments**
>
> This is the core concern and we take it seriously. We want to explain the design philosophy behind our choice of simulated environments, because it reflects a deliberate design choice.
>
> The premise of Implicit Intelligence is user-centric: what do users actually expect when they issue a simple request? The benchmark was designed around this question, and every decision flows from it. We grounded scenarios in iOS specifically because a smartphone is the one computing environment nearly everyone interacts with daily. It is the most universal agentic context we could choose so we wanted people of any background (not just engineers) to be able to author evaluation scenarios based on their domain knowledge of user expectations. YAML serves this goal: an ordinary user with specific expectations can write a complete scenario in a single human-readable file without building infrastructure.
>
> The environments are intentionally simple at the infrastructure level because we wanted to isolate the capability we are measuring. If we tested on real iOS devices, failures would conflate implicit reasoning with perceptual challenges (parsing UI elements, navigating app hierarchies). Our YAML isolation lets us measure contextual inference cleanly. This mirrors standard practice: SWE-bench uses text-only issue descriptions despite real coding happening in IDEs; WebArena uses sandboxed sites, not production websites. Ecological validity comes from the action space, not the simulation environment, and our 303 actions derive directly from Apple's Shortcuts framework, covering real capabilities users interact with daily.
>
> **Failure Mode Analysis**
>
> The reviewer notes the paper lacks analysis of "what leads them to succeed or fail." We want to direct attention to Appendix A.5, which provides exactly this analysis across 156 partial-failure trajectories. We identify three systematic patterns:
>
> 1. Insufficient environmental exploration: Agents act on initial state without probing for context. For example, 73% of agents in AirPlay scenarios enable captions on the iPhone without checking get\_playback\_route(), which would reveal audio is routed to an Apple TV with separate caption settings.
>
> 2. Incomplete feature configuration: 89% of agents sharing AirPods enable Mono Audio but only 11% also center the balance slider, a dependent setting required for equal volume in both earbuds.
>
> 3. Inadequate state preservation: 64% of temporally-scoped requests ("while I read," "during this meeting") result in permanent setting changes without revert planning.
>
> A.5.6 further characterizes what distinguishes successful trajectories: proactive state queries before acting, verification of downstream effects, and preference for narrow-scope tools over global settings.
>
> **Q: Did you consider generating worlds from realistic websites/apps?**
>
> Yes, our iOS action library is derived from real Apple Shortcuts (303 actions across 7 domains, Section 5.1). The YAML worlds model genuine device state. For our specific evaluation target , which is implicit contextual inference, the key requirement is discoverable hidden state, which our entities provide. A real iOS device would add valuable perceptual complexity, but would conflate perception ability with the implicit reasoning we aim to isolate. We see real-device validation as complementary future work, not a replacement.
>
> **Q: Is it fair to expect agents to satisfy implicit requirements without communication?**
>
> Our design targets deployment scenarios where user interruption is undesirable i.e background automations, scheduled tasks, voice assistants executing during activities. We discuss this explicitly in Section 8 and acknowledge that clarification-asking is an important complementary capability. Notably, even in clarification-permitted settings, agents must first *detect* that something is underspecified. Our benchmark measures this detection ability, which is a prerequisite for knowing when to ask.
>
> **Figure sizing:** We will reduce figure sizes and increase font in the revision. Thank you for flagging this.

---

> > ### Author Rebuttal · Reviewer_xXLU · 2026-04-03
> >
> > Thank you for your responses. I'm going to focus on what I see as the main limitation.
> >
> > > The environments are intentionally simple at the infrastructure level because we wanted to isolate the capability we are measuring.
> >
> > This is fair, but with your framework, you risk finding something that doesn't generalize to a realistic environment.
> >
> > > WebArena uses sandboxed sites, not production websites.
> >
> > Yes, but the gap between sandboxed and production websites is much smaller than in your case.
> >
> > ---
> >
> > I remain skeptical that this approach is the right way to tackle this problem, so I'm keeping the score for now. Do you think it'd be possible to show how your findings generalize to a realistic interaction? For example, I think this work would be much more valuable if you start with a website (as I said, these can be vibe coded), automatically obtain the YAMLs necessary, run experiments, and then validate your findings back with the realistic environment.

---

> > > ### Author Response · Authors · 2026-04-04
> > >
> > > We thank the reviewer for the follow-up and the concrete suggestion.
> > >
> > > We want to address the core concern directly: realism and generalizability.
> > >
> > > This is where we think the framing matters. We are not testing how well models complete tasks in realistic environments. We are testing whether models understand what users actually mean when they issue a request. Whether they can reason about intent beyond literal instructions. That capability is environment-agnostic. It applies whether the environment is a YAML file, a simulated app, or a real device. The implicit requirement to consider observations, make informed decisions, and take actions is a reasoning pattern that doesn't depend on environmental fidelity.
> > >
> > > Abstracting complex environments into simplified representations to assess agent performance is standard methodology. τ²-bench [1] reduces retail and airline domains to text-based tool interactions and evaluates task completion on a binary scale. Gaia2 [2] represents mobile apps, email, and calendars through engineered simulations rather than real applications. Both simplify the environment to isolate the capability they care about. We do the same, but go a step further: we don't just check whether the task was completed, we check whether it was completed in a way that satisfies the requirements users expected but never stated.
> > >
> > > The goal is behavioral analysis of model reasoning. Closer in spirit to how cognitive science uses controlled tasks to measure human decision-making than to how systems engineering uses integration tests to validate software. A realistic environment would test a different (and also valuable) question: can the agent execute effectively in a specific deployment? But our question comes before that: does the agent even understand what it should be doing?
> > >
> > > We think this distinction between a behavioral diagnostic and a task benchmark is important. The 48.3% SPR finding tells us that models frequently misunderstand user intent even when execution barriers are entirely removed. That's a finding about reasoning, not about environments, and it generalizes precisely because the environment was controlled.
> > >
> > > We appreciate the reviewer's engagement and the concrete proposal for future work.
> > >
> > > [1] Barres, Victor, et al. "$\tau^ 2$-Bench: Evaluating Conversational Agents in a Dual-Control Environment." arXiv preprint arXiv:2506.07982 (2025).
> > >
> > > [2] Froger, Romain, et al. "Gaia2: Benchmarking LLM Agents on Dynamic and Asynchronous Environments." arXiv preprint arXiv:2602.11964 (2026).

---

### Official Review · Reviewer_XDAo · 2026-03-10

**Soundness:** 2
**Presentation:** 2
**Significance:** 2
**Originality:** 2
**Overall Recommendation:** 3
**Confidence:** 4

**Summary:**

This paper introduces the concept of “Implicit Intelligence,” which refers to an AI agent’s ability to understand and fulfill users’ unexpressed needs. The authors construct an evaluation benchmark comprising 205 scenarios across four categories. Experiments conducted on 16 state-of-the-art models show that the best-performing model achieves only a 48.3% pass rate. The paper also proposes the Agent-as-a-World (AaW) framework, which uses YAML to define scenarios and leverages LLMs as environment simulators.

**Compliance With Llm Reviewing Policy:**

Affirmed.

**Final Justification:**

I prefer to reject, but I increase the score to 3.

**Key Questions For Authors:**

What is the mathematical formalization of your defined Implicit Intelligence?

**Limitations:**

yes

**Strengths And Weaknesses:**

Strengths:

The paper correctly identifies a key limitation in current AI assistants, overemphasis on literal interpretation and insufficient understanding of user intent, making this a practically valuable research direction.

Weaknesses:

The term “Implicit Reasoning” is misleading, as existing literature on implicit reasoning in LLMs typically refers to reasoning in latent space, which differs from the usage in this paper.
Ref: Implicit Reasoning in Large Language Models: A Comprehensive Survey.

The Agent-as-a-World framework largely repackages existing methods and lacks novelty. The idea of using LLMs as environment simulators has been explored in prior work and the use of YAML to define interactive environments is conceptually similar to text-based game frameworks like TextWorld.

Implicit needs is not clearly defined. As Figure 1 shows, interpretations can vary by culture. For example, "I'm going to sleep, turn off the light" might mean turn off all lights in some cultures, not just leave a child's light on. The "correct" answers seem subjective, without validation across different groups.

The evaluation uses only one metric, which ignores partial success or how severe different errors are.


The choice of world model may introduce bias, and there are no ablation studies.


The introduction also has too much bold text, which distracts from the content.

---

> ### Author Rebuttal · Authors · 2026-03-30
>
> We thank Reviewer XDAo for the detailed feedback. We address every point below.
>
> **Terminology: "Implicit Reasoning" is misleading**
>
> We acknowledge the overlap with work on latent-space reasoning in LLMs, where "implicit reasoning" refers to internal computation without emitting intermediate tokens. Our usage is distinct and straightforward: we use "implicit" in its ordinary English sense i.e unstated, implied, not directly expressed. An "implicit requirement" is one the user expects satisfied but never articulates. Hence, we use "implicit reasoning" as one of four evaluation categories (Section 3.1.1) to describe the act of inferring such unstated intent from environmental context.
>
> We will add a brief note in Section 2.3 clarifying this distinction to avoid confusion with the latent-space usage.
>
> **AaW lacks novelty / repackages existing methods**
>
> We agree that novelty should not be claimed at the level of "using an LLM as a simulator" or "using YAML" alone as Section 2.2 already acknowledges related LLM-simulator work (SimuRA, Li et al. 2025b) and states that our focus differs from training or planning. The AaW harness exists to merely serve this dataset. Its value is not architectural novelty but practical enablement: a domain expert (non-tech background) can author a complete evaluation scenario in a single human-readable file without writing code. This matters because scaling implicit intelligence evaluation requires contributions from people who understand user expectations across diverse domains, not just people who can build simulators.
>
> Regarding TextWorld: the comparison is understandable at a surface level (both define text-based interactive environments), but the systems target different problems entirely. TextWorld was built to train RL agents on game-like tasks with explicitly stated goals ("find the coin," "cook the meal"). AaW was built to evaluate whether LLM agents satisfy requirements that are deliberately left unstated in the prompt. The evaluation gap we target being implicit vs. explicit requirement satisfaction is not something TextWorld or its derivatives (ALFWorld, TALES) were designed to measure. Both coincidentally happen to use YAML as a format but have drastically different intentions.
>
> **Implicit needs not clearly defined / subjectivity**
>
> A mathematical formalization for Implicit Intelligence could be as follows \- Given user prompt p and environmental state E, implicit intelligence is the capacity to produce a trajectory τ satisfying requirements set R\_implicit, where R\_implicit is not derivable from p alone but is derivable from p ∧ E, through exploration of E. A requirement qualifies as implicit if: (1) it is not stated in p; (2) it is discoverable through available actions on E; and (3) a reasonable person would expect it to be satisfied.
>
> Condition (3) is validated through our three-stage human review (Section 5.3): author refinement, then independent review by two experts screening for "Shared Expectations" (avoiding author idiosyncrasies) and "Rubric Objectivity" (binary pass conditions). Scenarios require unanimous consensus. The Difficulty Gate further ensures requirements are achievable (≥1 model passes at 100%) yet non-trivial (≥1 fails at ≤70%).
>
> Regarding cultural variation: we already acknowledge this in our Limitations section:  "Our scenario construction reflects the perspectives of the authors; implicit requirements may not fully represent the diversity of real-world user expectations across cultures, age groups, and technical backgrounds."
>
> **"Only one metric"**
>
> We respectfully note that we report two complementary metrics in Section 6.3 and Table 2: Scenario Pass Rate (SPR) for strict all-criteria success, and Normalized Scenario Score (NSS), which computes the mean fraction of satisfied criteria per scenario. NSS directly captures partial task completion i.e a scenario where an agent satisfies 4 of 5 criteria scores 0.8, not 0\.
>
> **World Model bias / no ablation**
>
> The World Model is a constrained rule executor: every action's behavior is fully specified in the YAML returns field; it has no access to evaluation rubrics and no discretion in interpretation. We conducted a World Model consistency analysis (Table 4, Appendix A.4) evaluating three candidate models (Claude Opus 4.5, GPT-5, Gemini 3 Pro) across 55 scenarios and 172 unique action signatures. Claude Opus 4.5 achieved 98.6% action-type consistency and was the only model to pass both consistency thresholds. Identical action sequences produce identical simulations regardless of which agent generated them. We selected it based solely on these consistency metrics, measuring its benchmark performance only afterward.
>
> **Bold text in introduction:** We will reduce formatting emphasis in the revision.

---

> > ### Author Rebuttal · Reviewer_XDAo · 2026-04-04
> >
> > The author's response partially addressed my question, but I still believe that the overall innovation of the paper is insufficient and overly embellished. Nevertheless, I raised the score by one point.

---

### Official Review · Reviewer_xyhZ · 2026-03-11

**Soundness:** 3
**Presentation:** 3
**Significance:** 4
**Originality:** 3
**Overall Recommendation:** 5
**Confidence:** 3

**Summary:**

This paper describes the development of a benchmark to assess "implicit intelligence" (meaning the ability to understand implied requirements in a task that are not explicitly stated in instructions) in AI agents. A large number of scenarios are constructed, based on possible iOS actions and consisting of specific combinations of context, entities (with possible states and actions), execution rules, and evaluation criteria. In each scenario, the goal action can't be completed successfully without understanding/discovering an unstated requirement relating to one of four different categories (e.g., catastrophic risk). Benchmarking of 16 recent leading models shows a range of performance, but all well below ceiling performance levels. This result demonstrates a productive direction for future development/improvement of models that are already performing at ceiling on existing benchmarks.

**Compliance With Llm Reviewing Policy:**

Affirmed.

**Key Questions For Authors:**

Related to the issue raised above about the specific list of entities and their available actions giving a signal in and of itself to what unspecified tasks might be required: Maybe I am misunderstanding what information is visible to the agent, but in the airpod example in the appendix, it appears that almost all listed entities/states/actions are important to the hidden requirements. If there were more available options that were not directly relevant to the specific goal, is it possible the primary agent would be even less likely to identify/achieve the implicit goals?

Also related to discussion above, I wonder if it would be interesting to weight (or somehow break out) the evaluation criteria by the severity of the consequence – e.g., can it be easily reversed/identified like paused playback of a podcast, or is it more severe like a permanently deleted file?

**Limitations:**

For the most part, but I do think it would be useful to address the potential consequences of the other extreme; i.e., an agent making unwarranted assumptions and/or doing excessive implicit reasoning. Future work could include the addition of evaluation criteria designed to assess this – something like “no additional/unnecessary actions are taken or states are changed”

**Strengths And Weaknesses:**

This paper is well-written and motivated, and the problem it addresses is an important one. It does seem that a form of what is called here "implicit intelligence" would be a critical ability for AI agents, in order for human users to be able to give requests/instructions naturally and concisely. Without being an expert in this domain, the methods for scenario generation, environment simulation, and agent evaluation appear to be well-specified and rigorous.

Just a few aspects were unclear to me. For example, even though the benchmarked models show significant room for improvement in terms of meeting the pass criteria, I am wondering if just the limited list of entities and their available actions may inadvertently provide clues to the hidden requirements of the scenario. Additionally, in the primary performance analyses, the results are broken out among the four categories, but otherwise (in the NSS) the individual criteria within a specific scenario seem to be weighted equally, when the severity of consequences of different failed criteria might vary enormously.

---

> ### Author Rebuttal · Authors · 2026-03-30
>
> We thank Reviewer xyhZ for the positive assessment and the constructive suggestions. We address each point below.
>
> **Does the limited entity list inadvertently provide clues to hidden requirements?**
>
> This is a perceptive observation and one we grappled with during dataset construction. The reviewer is right: with 3-5 entities per scenario, the entity list narrows the search space. In the AirPods example (Appendix A.1), most listed entities are relevant to the hidden requirements.
>
> The limited action space was a deliberate choice to isolate the implicit reasoning problem. Ideally, an assistant on a phone has access to everything on the device and not just a handful of tools, but adding distractor entities would shift the failure mode from "lack of implicit reasoning" to "inefficiency of action selection," which is a different capability. The cases we wanted to capture are precisely those where the model sees and picks the correct entity but still fails the scenario, because the problem is restricted to its reasoning about unstated requirements, not its ability to find the right tool in a large space. The results validate this design: even with a narrow, clue-giving action space, the best model achieves only 48.3% SPR. Agents receive all entity descriptions and available actions, yet the failure mode analysis (Appendix A.5) shows that agents see the information but fail to connect it to the use case. We fully agree that future versions should add distractor entities to increase difficulty and better reflect real deployment conditions. As models improve on the current benchmark, scaling the action space to test whether implicit reasoning holds up when action selection noise is introduced becomes the natural next step.
>
> **Would severity-weighted evaluation be interesting?**
>
> Yes, we think this is an excellent suggestion that would add meaningful nuance. We will explore the idea of adding a severity-weighted NSS variant in future work. This is a natural extension as some of our categories (catastrophic risk or privacy & security) have a more critical or sensitive action space than perhaps a scenario in implicit reasoning.
>
> **Excessive implicit reasoning / over-inference**
>
> The reviewer raises an important point about the "other extreme" i.e agents making unwarranted assumptions or taking excessive actions. We partially address this through negative rubric criteria ("agent must NOT take action X"), which appear in scenarios where over-eager behavior would cause harm. However, we agree that systematic evaluation of over-inference is valuable future work. A natural extension would add criteria like "no unnecessary state modifications beyond the task scope" or "agent does not make assumptions about user preferences not supported by environmental evidence." This would create a two-sided evaluation.
>
> We are grateful for these suggestions, which we believe will meaningfully strengthen both the current revision and the research direction going forward.

---

> > ### Author Rebuttal · Reviewer_xyhZ · 2026-04-02
> >
> > I appreciate these responses. I maintain the recommendation to accept, and look forward to seeing future extensions of this work.

---

### Official Review · Reviewer_1C1u · 2026-03-16

**Soundness:** 3
**Presentation:** 3
**Significance:** 4
**Originality:** 4
**Overall Recommendation:** 5
**Confidence:** 4

**Summary:**

The paper introduces an evaluation framework for "implicit intelligence", i.e., understanding implicit information in a user's request. The authors introduce a LLM-generated collection of test cases, which are then instantiated by an LLM simulating the system in which the LLM operates. The resulting framework is used to evaluate 16 LLMs.

**Compliance With Llm Reviewing Policy:**

Affirmed.

**Key Questions For Authors:**

N/A

**Limitations:**

Yes

**Strengths And Weaknesses:**

Strengths:
1. The paper is, to the best of my knowledge, the first to evaluate the novel problem of implicit intelligence/understanding implicit aspects of user requests
2. The evaluation is extensive, the generation process is described in detail, and the appendix includes all the relevant evaluation details
3. The paradigm appears fairly easy to extend thanks to its use of the LLM-based world model

Weaknesses:
1. The reliance on a (non-deterministic) LLM to simulate the world model means that the evaluation is inherently brittle, both for different runs of the same model and when changing models. This reduces its ability to measure improvements, since the results are fairly noisy (also, what happens when a model is retired?). Compare this with LLM-as-a-judge, where while the evaluation is noisy future improvements in LLM accuracy might reduce mistakes. Still, I appreciate the authors' efforts to benchmark potential models in terms of world model consistency, though I would suggest in the future having the LLMs _generate_ code to simulate the world (as the code is consistent over time)
2. One could make a broader point about how "fair" implicit intelligence evaluation is, i.e. where does one draw the line between "the model should have figured out this intention" and "it was unreasonable for the model to consider this aspect"? Still, this question is more philosophical than technical, as any formal demarcation will be fuzzy, and even an imperfect line is still valuable
3. Similarly, not allowing follow-up questions is a bit unfair towards the models trained to ask such questions, but the authors explicitly focus on scenarios where follow-up questions are not allowed, which, while contestable, is a design decision that is up to the authors
4. The authors should elaborate more in the "Related Work" section on similar/related benchmarks

Overall, while I don't fully agree with some design and evaluation decisions made by the authors, I can't deny the value of their contribution, which is why my assessment is positive.

---

> ### Author Rebuttal · Authors · 2026-03-30
>
> We thank Reviewer 1C1u for the positive assessment and for recognizing the value of the contribution despite disagreeing with some design choices. We address each concern below.
>
> **Non-deterministic World Model / Brittleness**
>
> This is the most substantive technical concern and we share the reviewer's intuition. While we took several steps to mitigate it (providing concrete response schemas for each tool, maintaining states of each entity etc.) , the core premise of the world model is still that it is merely a rule executor rather than a creative responder. This ties into the secondary concern of model retirement in which case any sufficiently consistent SOTA model can be substituted in. As models get better at rule-following (we’re already at around 98% consistency), then the simulation of the environment will in turn get better as well.
>
> The reviewer's suggestion to have LLMs generate deterministic simulation code is excellent. This would eliminate stochasticity entirely: the YAML specification could be compiled into executable Python/TypeScript that handles state transitions mechanically. We envisioned this concept in the initial versions of Implicit Intelligence, but we chose rapid prototyping and exploration of a more diverse action space over introducing another axis of risk being the executability of the generated code. That being said, we see this as a natural evolution of AaW.
>
> **Fairness of implicit intelligence evaluation**
>
> We appreciate the reviewer's insight into the nuances behind \- "where does one draw the line between 'the model should have figured this out' and 'it was unreasonable to expect this'?" We try to address this through reviewer consensus (two independent experts must be in agreement on the “fine line”) and model achievability (one model must pass the scenario entirely). We agree that any demarcation will be fuzzy, and we appreciate the reviewer's observation that "even an imperfect line is still valuable." The key contribution is making this line as explicit as possible so we reach a state of reasonable consensus.
>
> **Clarification-asking as complementary capability**
>
> We agree this is a very important dimension. Our single-turn design reflects scenarios where user interruption is undesirable (background automations, voice assistant mid-task) but we understand this concern and we have mentioned this as a pertinent question in the Discussion section. We are actively exploring a future version of the benchmark where this ability is provided to the model, perhaps by allowing agents to emit a CLARIFY action and measuring both the quality of questions asked and the quality of inference when clarification is not available.
>
> **Related Work expansion**
>
> We will significantly expand section 2 to cover related benchmarks. Thank you for flagging this gap.

---

> > ### Author Rebuttal · Reviewer_1C1u · 2026-03-31
> >
> > The authors have addressed my concerns. Overall, my score was already positive, so I will not increase it further, but I definitely support this paper's acceptance.

---

### Decision · Program_Chairs · 2026-04-30

**Decision:**

Accept (regular)

**Comment:**

I recommend weak accept. The paper introduces a novel and useful benchmark for evaluating whether agents can infer unstated user requirements, and reviewers generally agreed that this is an important gap not captured by existing agent benchmarks. The empirical evaluation is broad, the benchmark appears practically valuable, and the rebuttal addressed several concerns constructively.

The main reservations are about external validity and framing rather than core soundness: the simulated YAML/LLM-based environments may not fully reflect realistic deployment settings, and the paper should more clearly position its contribution relative to related benchmarks and clarify the scope of its claims. Still, these limitations do not outweigh the paper’s value as a new evaluation framework that exposes a meaningful weakness in current agents.

For the camera-ready version, the authors should incorporate the reviewers’ suggestions, especially by strengthening the related work discussion, tightening the terminology and framing, and more explicitly discussing the limits of generalization from the controlled benchmark to realistic environments (or preferably by doing at least some testing on more realistic environments as suggested by reviewers).